# Faster Algorithms and Constant Lower Bounds for the Worst-Case Expected Error

**Jonah Brown-Cohen**
Chalmers University of Technology
`jonahb@chalmers.se`

## Abstract

The study of statistical estimation without distributional assumptions on data values, but with knowledge of data collection methods was recently introduced by Chen, Valiant and Valiant (NeurIPS 2020). In this framework, the goal is to design estimators that minimize the worst-case expected error. Here the expectation is over a known, randomized data collection process from some population, and the data values corresponding to each element of the population are assumed to be worst-case. Chen, Valiant and Valiant show that, when data values are $\ell_\infty$-normalized, there is a polynomial time algorithm to compute an estimator for the mean with worst-case expected error that is within a factor $\frac{\pi}{2}$ of the optimum within the natural class of semilinear estimators. However, their algorithm is based on optimizing a somewhat complex concave objective function over a constrained set of positive semidefinite matrices, and thus does not come with explicit runtime guarantees beyond being polynomial time in the input. In this paper we design provably efficient algorithms for approximating the optimal semilinear estimator based on online convex optimization. In the setting where data values are $\ell_\infty$-normalized, our algorithm achieves a $\frac{\pi}{2}$-approximation by iteratively solving a sequence of standard SDPs. When data values are $\ell_2$-normalized, our algorithm iteratively computes the top eigenvector of a sequence of matrices, and does not lose any multiplicative approximation factor. Further, using experiments in settings where sample membership is correlated with data values (e.g. "importance sampling" and "snowball sampling"), we show that our $\ell_2$-normalized algorithm gives a similar advantage over standard estimators as the original $\ell_\infty$-normalized algorithm of Chen, Valiant and Valiant, but with much lower computational complexity. We complement these positive results by stating a simple combinatorial condition which, if satisfied by a data collection process, implies that any (not necessarily semilinear) estimator for the mean has constant worst-case expected error.

## 1 Introduction

Standard methods in statistical analysis are often based on the assumption that data values are drawn independently from some underlying distribution, and may perform poorly when this assumption is violated. Methods from robust statistics often modify this assumption, for example by allowing a small fraction of data values to be arbitrary outliers while the bulk of the data values remain independent samples. However, there are natural settings in which data values may be strongly correlated to sample membership, so that distributional assumptions (e.g. independence of data values) can lead to inaccurate results. Furthermore, inaccuracy of predictions in such settings can have serious societal consequences–for example predictions of election outcomes or the spread of a contagious disease.

35th Conference on Neural Information Processing Systems (NeurIPS 2021).

Recently, Chen, Valiant, and Valiant [4] introduced a framework to capture statistical estimation in such difficult settings, by assuming that the data values are worst-case, but that the estimation algorithm is able to leverage knowledge of the randomized data collection process. The framework is simple to describe. There is a set $\{1, \ldots n\}$ of $n$ indices with corresponding data values $x = \{x_1, \ldots x_n\}$ along with a probability distribution $P$ over pairs of subsets $A, B \subseteq \{1, \ldots n\}$. Here $A$ is called the sample set and $B$ is called the target set. A sample $(A, B)$ is drawn from $P$ and the values $x_A = \{x_i \mid i \in A\}$ are revealed. The goal is then to estimate the value of some function $g(x_B)$ given only $A$, $B$ and $x_A$. The quality of an estimator in this framework is measured by its *worst-case expected error*. Here the data values $x$ are worst-case and the expectation is with respect to the distribution $P$.

The worst-case expected error framework captures several natural settings where data values may be correlated to sample membership. *Importance sampling* is the setting where the target set $B$ is always equal to the whole population $\{1, \ldots n\}$, and the sample set $A$ is chosen by independently sampling each element $i \in \{1, \ldots n\}$ with probability $p_i$. In the worst-case expected error framework, the values $x_i$ may be arbitrarily correlated to the probability $p_i$ of being sampled. In *snowball sampling* [12], the elements of $\{1, \ldots n\}$ correspond to the vertices of a graph (e.g. a social network). A sample set $A$ is chosen by first picking a few initial vertices. Next, each chosen vertex recruits a randomly chosen subset of its neighbors into the sample, and this process is repeated until a desired sample size is reached. The target set $B$ can be either the whole population or a subset given by a few additional iterations of the recruitment process. Here it is natural to assume that the data values $x_i$ at neighboring vertices will be highly correlated. Finally, *selective prediction*[10, 18] is a temporal sampling method where the population $\{1, \ldots n\}$ corresponds to time steps, and the goal is to predict the average of future data values given the past (e.g. the average change in the stock market). The target set $B$ is some time window $\{t, \ldots t + w\}$ and the sample $A$ is $\{1, \ldots t\}$. If the random process for choosing the starting point $t$ and length $w$ of the time window is chosen appropriately, sub-constant error is attainable even when the $x_i$ are chosen to be worst-case values bounded by a constant[10, 18].

In [4], the authors design an algorithm for computing estimators for the mean of the target set $B$. In particular, they restrict their attention to the class of *semilinear* estimators i.e. estimators which are restricted to be a linear combination of the sample values $x_A$, but where the weights of the linear combination may depend arbitrarily on $P$, $A$, and $B$. In the setting where the data values are bounded, the authors design an algorithm that outputs a semilinear estimator with worst-case expected error at most a $\frac{\pi}{2}$ factor larger than that of the optimal semilinear estimator. They then demonstrate that the estimator output by their algorithm has significantly lower expected error than many standard estimators in the importance sampling, snowball sampling, and selective prediction settings where data values are correlated to sample membership. However, several open questions remain, even for the case of computing the optimal semilinear estimator for the mean.

First, the algorithm in [4] is a concave maximization over a constrained set of positive semidefinite matrices, which is solved using a general purpose convex programming solver. In particular this means that no explicit bounds on the runtime of the algorithm are given beyond being polynomial in $n$. This leads us to ask:

> *Question 1: Is there a simpler algorithm to compute a semilinear estimator with approximately optimal worst-case expected error that comes with explicit runtime guarantees?*

Second, the choice of $\ell_\infty$-normalization (i.e. assuming the data values are bounded) is natural for some settings such as polling or testing for infectious disease. However, $\ell_2$-normalization (i.e. $\frac{1}{n} \sum_i x_i^2 \leq 1$) is also a natural choice for settings where some data values may be much larger than others such as predicting stock prices. This motivates:

> *Question 2: Is there an efficient algorithm to compute a semilinear estimator with approximately optimal worst-case expected error in the $\ell_2$-bounded case?*

Finally, the setting of the worst-case expected error is quite challenging, and so one would expect that for many randomized data collection processes it is not possible to achieve non-trivial worst-case expected error. Thus, one might ask:

> *Question 3: Are there simple properties of a data collection process that ensure that no estimator can achieve sub-constant worst-case expected error?*

## 1.1 Our Results

We answer the first two questions by designing simple algorithms based on online gradient descent for approximating the optimal semilinear estimator in both the $\ell_\infty$-bounded and the $\ell_2$-bounded case. To be able to describe our algorithms we first formally define the input.

**Definition 1.1** (Definition 2 in [4]). A joint sample-target distribution $P$ over $\{1, \ldots n\}$ is given by the uniform distribution over $m$ pairs $(A_i, B_i)$ where $A_i, B_i \subseteq \{1, \ldots n\}$.

Because any distribution $P$ on pairs $(A, B)$ may be approximated to arbitrary accuracy by a uniform distribution this is a reasonable choice of parametrization. In many settings the $m$ uniform pairs are obtained by sampling sufficiently many times from the known randomized data-collection process (see supplementary material for details). Thus, one should think of $m$ as being much larger than $n$, say at least $\Omega(n^2)$. Our algorithms' runtimes also depend on a parameter $\rho$ which we define to be $n$ times the worst-case expected error of the optimal semilinear estimator.

**Theorem 1.2** (Informal–see Theorem 2.3). *There is an algorithm based on online gradient descent for approximating the optimal semilinear estimator to within a multiplicative $\frac{\pi}{2}$ and additive $\epsilon$ error for $\ell_\infty$-bounded values. The algorithm runs in $O(\rho^2 \log \rho)$ iterations each taking $\widetilde{O}(mn^{\omega-1} + n^{7/2})$ time, where $\omega$ is the current matrix multiplication exponent.*

Note that when there exists a semilinear estimator achieving worst-case expected error $O(\frac{1}{n})$, the algorithm requires $O(1)$ iterations. Each iteration of the algorithm requires solving a standard SDP in $n$ dimensions with $n$ constraints (in fact the only constraints are that the diagonal entries of the psd matrix are constrained to all be equal to one).

**Theorem 1.3** (Informal–see Theorem 2.4). *There is an algorithm based on online gradient descent for approximating the optimal semilinear estimator to within an additive $\epsilon$ error for $\ell_2$-bounded values. The algorithm runs in $O(\rho^2 \log \rho)$ iterations each taking $O(mn)$ time.*

In addition to the improved runtime, each iteration of the algorithm for $\ell_2$-bounded values only requires computing the eigenvector corresponding to the largest eigenvalue of an $n \times n$ matrix, making the algorithm practical and simple to implement. We provide the details of both algorithms in Section 2.

Recall that the SDP algorithm from [4] obtained lower expected error than several standard estimators on various synthetic datasets where data values are correlated to sample membership. To compare to these empirical results obtained for the $\ell_\infty$-bounded case, we implement our algorithm for $\ell_2$-bounded values and perform experiments on the same synthetic datasets from [4]. We find that the expected error of our algorithm on these synthetic datasets matches that of the SDP algorithm from [4], while being simpler to implement and having lower computational complexity. See Section 5 for details.

Finally, in Section 4 we answer the third question by providing a simple combinatorial condition for a sample-target distribution $P$, which ensures that any (not necessarily semilinear) estimator has constant worst-case expected error.

## 1.2 Related Work

The most closely related work is [4] which defined the framework for the worst-case expected error and designed the first algorithms in this setting. The papers [10, 18] on selective prediction are also closely related as they design efficient estimators in the worst-case expected error setting for a specific choice of data collection distribution.

There has been extensive recent work on the theory and practice of robust estimation in high dimensions, where some fraction of data values are arbitrarily corrupted and the rest remain independent samples from the distribution [5, 16, 3, 6, 7, 8]. Most closely related to our work is the recent paper of Hopkins, Li and Zhang on robust mean estimation using online convex optimization/regret minimization [13]. While that work focuses on a quite different problem setting, the high-level idea of using methods in online convex optimization to design simple and efficient algorithms for robust estimation is closely connected to our work.

## 1.3 Preliminaries

We focus on the setting where the data values $x$ are real numbers and the goal is to estimate the mean of the target set. Since the mean scales linearly with the data values, it is natural to assume that the values $x$ lie in some bounded set $D \subseteq \mathbb{R}^n$. For our purposes $D$ will either be the $\ell_\infty$-ball or the $\ell_2$-ball.

**Definition 1.4.** Given a sample-target distribution $P$ and a bounded set $D \subseteq \mathbb{R}^n$, an estimator $f(x_A, A, B)$ is a real-valued function which takes as input the data values $x_A$ and the sample-target index sets $(A, B)$. The *worst-case expected error* of $f$ on $P$ is given by

$$\max_{x \in D} \frac{1}{m} \sum_{i=1}^{m} \left( f(x_{A_i}, A_i, B_i) - \text{mean}(x_{B_i}) \right)^2$$

We use $\|v\|$ to denote the $\ell_2$-norm of a vector $v$ and $\|v\|_\infty$ to denote the $\ell_\infty$-norm. For a matrix $M$ we use $\|M\|$ to denote the operator norm. For a subspace $W \subseteq \mathbb{R}^n$, we use $W^\perp$ to denote the orthogonal complement subspace, and we write $\Pi_W$ for the orthogonal projection onto $W$.

## 2 Algorithms for the Worst-case Expected Error

In this section we design algorithms for approximating the best semilinear estimator for the mean using tools from online convex optimization. As in [4] we focus on the class of semilinear estimators which compute a linear combination of the data values $x_A$ where the weights of the linear combination can depend on the sets $A$ and $B$. For convenience we introduce the notation $W_i \subseteq \mathbb{R}^n$ to denote the subspace of vectors that have non-zero coordinates only on indices in $A_i$. Then a semilinear estimator takes the form $f(x_{A_i}, A_i, B_i) = \langle a_i, x \rangle$ where each vector $a_i \in W_i$. To further simplify notation we denote by $b_i \in \mathbb{R}^n$ the vector which takes value $\frac{1}{|B_i|}$ on coordinates in $B_i$ and is zero otherwise, so that $\text{mean}(x_{B_i}) = \langle b_i, x \rangle$. Using this notation we have:

**Definition 2.1.** Given a sample-target distribution $P$ and a set $D \subseteq \mathbb{R}^n$ the worst-case expected error of the optimal semilinear estimator is given by

$$\text{opt}(P) = \min_{\{a_i \in W_i\}_{i=1}^m} \max_{x \in D} \frac{1}{m} \sum_{i=1}^{m} \langle a_i - b_i, x \rangle^2$$

**SDP Relaxation.** The first step to approximating the best semilinear estimator is to introduce an SDP relaxation for the inner maximization in Definition 2.1. To simplify notation we make the following definition.

**Definition 2.2.** Let $\boldsymbol{a} = \{a_1, \ldots a_m\}$ where $a_i \in W_i$ for all $i$. Define

$$M(\boldsymbol{a}) = \frac{1}{m} \sum_i (a_i - b_i)(a_i - b_i)^\top.$$

Then the inner maximization in Definition 2.1 can be written more simply as

$$\max_{x \in D} x^\top M(\boldsymbol{a}) x$$

When $D = \{x \in \mathbb{R}^n \mid \|x\|_\infty \leq 1\}$ the inner maximization can be relaxed to the semidefinite program

$$\text{sdp}_\infty(\boldsymbol{a}) = \max_{X \succeq 0, X_{j,j}=1} \langle M(\boldsymbol{a}), X \rangle \tag{1}$$

which has an optimal value within a factor $\frac{\pi}{2}$ of the true worst-case expected error as shown via the rounding method of [17]. See [4] for a self-contained explanation. Therefore to approximate the best semilinear estimator it suffices to approximately solve the min-max optimization problem

$$\min_{\boldsymbol{a}} \text{sdp}_\infty(\boldsymbol{a}) \tag{2}$$

**Algorithm via Online Gradient Descent.** Our first theorem states that we can approximately minimize $\mathrm{sdp}_\infty(\boldsymbol{a})$ with an application of online gradient descent.

**Theorem 2.3.** *Let $P$ be a sample-target distribution, $\rho = n \cdot \mathrm{opt}(P)$, and $\epsilon > 0$. There is an algorithm which computes a set of $m$ vectors $\boldsymbol{a}' = \{a_1', \ldots a_m'\}$ such that*

$$\mathrm{sdp}_\infty(\boldsymbol{a}') \le \min_{\boldsymbol{a}} \mathrm{sdp}_\infty(\boldsymbol{a}) + \epsilon.$$

*The algorithm runs in $O\left(\frac{\rho^2 \log \rho}{\epsilon^2}\right)$ iterations, and $\widetilde{O}(\frac{\rho^2 \log \rho}{\epsilon^2}(mn^{\omega-1} + n^{7/2}\log(1/\epsilon))$ time, where $\omega$ is the current matrix multiplication exponent.*

Note that the parameter $\rho$ depends on the optimal value of the worst-case expected error. In particular, if the optimal worst-case expected error is $O(\frac{1}{n})$ then $\rho = O(1)$, and the iteration count of the algorithm is $O(\frac{1}{\epsilon^2})$. The algorithm is an application of online gradient descent where in each step we solve an instance of $\mathrm{sdp}_\infty(\boldsymbol{a})$ and use the solution as a convex cost function. The analysis is based on standard regret bounds for online gradient descent.

---

**Algorithm 1:** Online gradient descent for minimizing $\mathrm{sdp}_\infty(\boldsymbol{a})$.

**Input:** A sample-target distribution $P$, accuracy parameter $\epsilon$, upper bound on optimum $p$.
**Output:** A sequence of $m$ vectors $\boldsymbol{a} = \{a_1, \ldots a_m\}$ with $a_i \in W_i$ for all $i$.

1 Let $r = \sqrt{\frac{\pi m p}{2}}$
2 Let $\beta = \sum_{i=1}^m \|\Pi_{W_i^\perp} b_i\|^2$.
3 Let $a_i^{(1)} = \frac{1}{|A_i|}\mathbb{1}_{A_i}$ for all $i$.
4 **for** $t$ *from* 1 *to* $\frac{36\pi^2 n^2 p^2}{\epsilon^2}$ **do**
5 $\quad$ Let $\boldsymbol{a}^{(t)} = \{a_1^{(t)}, \ldots a_m^{(t)}\}$, and let $\eta_t = \frac{m}{n\sqrt{t}}$.
6 $\quad$ Let $X^{(t)}$ be a $\left(1 + \frac{\epsilon}{10}\right)$-approximate solution to $\mathrm{sdp}_\infty(\boldsymbol{a}^{(t)})$.
7 $\quad$ $a_i^{(t+1)} \leftarrow a_i^{(t)} - \eta_t \frac{2}{m}\Pi_{W_i}X^{(t)}(a_i^{(t)} - b_i)$ for all $i$.
8 $\quad$ Set $\lambda^{(t)} = \min\left\{1, \sqrt{\frac{r^2 - \beta}{\sum_{i=1}^m \|a_i^{(t+1)} - \Pi_{W_i}b_i\|^2}}\right\}$.
9 $\quad$ $a_i^{(t+1)} \leftarrow \lambda^{(t)}a_i^{(t+1)} + (1 - \lambda^{(t)})\Pi_{W_i}b_i$.
10 **return** $\boldsymbol{a}^{(t^*)}$ where $t^* = \mathrm{argmin}_t\langle M(\boldsymbol{a}^{(t)}), X^{(t)}\rangle$.

---

We now give the analysis of the algorithm using standard regret bounds for online gradient descent (see the text [11]). See the supplementary material for the referenced lemmas.

*Proof.* The main observation is that the algorithm is precisely online gradient descent where the convex cost function observed at each step is given by $f_t(\boldsymbol{a}) = \langle M(\boldsymbol{a}), X^{(t)}\rangle$. Indeed viewing $\boldsymbol{a}$ as a vector in $nm$ dimensions (one for each $a_i$) the component of the gradient $\nabla f_t(\boldsymbol{a})$ corresponding to $a_i$ is $\frac{2}{m}\Pi_{W_i}X^{(t)}(a_i - b_i)$. Using the positive semidefinite Grothendieck inequality, one can show that for the optimal solution $\boldsymbol{a}^*$ lies in $B_{r^*}(\boldsymbol{b})$, the $\ell_2$-ball of radius $r^* = \sqrt{\frac{\pi m \, \mathrm{opt}(P)}{2}}$ centered at $\boldsymbol{b} = \{b_1, \ldots, b_m\}$ (Lemma A.1). Thus given an upper bound $p \ge \mathrm{opt}(P)$ it is sufficient to limit the search to $B_r(\boldsymbol{b})$ for $r = \sqrt{\frac{\pi m p}{2}}$. The last lines of the loop are just projection onto this ball (Lemma A.3). Finally, for $\boldsymbol{a}$ in $B_r(\boldsymbol{b})$, a straightforward calculation (Lemma A.2) gives $\|\nabla f_t(\boldsymbol{a})\| \le \frac{2nr}{m}$.

Thus the algorithm is online gradient descent with feasible set diameter $D = 2r$ and gradients bounded by $G = \frac{2nr}{m}$. Thus by the textbook analysis of online gradient descent [11] we have

$$\frac{1}{T}\sum_{t=1}^T f_t(\boldsymbol{a}^{(t)}) - \min_{\boldsymbol{a}' \in B_r(\boldsymbol{b})} \frac{1}{T}\sum_{t=1}^T f_t(\boldsymbol{a}') \le \frac{3GD}{2\sqrt{T}} = \frac{3\pi n p}{\sqrt{T}}.$$

Letting $\boldsymbol{a}^* = \mathrm{argmin}_{\boldsymbol{a}} \mathrm{sdp}_\infty(\boldsymbol{a})$ (which we know is contained in $B_r(\boldsymbol{b})$) we have

$$\frac{1}{T}\sum_{t=1}^T f_t(\boldsymbol{a}^{(t)}) - \frac{1}{T}\sum_{t=1}^T f_t(\boldsymbol{a}^*) \le \frac{3\pi n p}{\sqrt{T}}.$$

Noting that $f_t(\boldsymbol{a}^*) = \langle M(\boldsymbol{a}^*), X^{(t)} \rangle \leq \mathrm{sdp}_\infty(\boldsymbol{a}^*)$ yields

$$\frac{1}{T} \sum_{t=1}^{T} f_t(\boldsymbol{a}^{(t)}) \leq \mathrm{sdp}_\infty(\boldsymbol{a}^*) + \frac{3\pi np}{\sqrt{T}} \leq \mathrm{sdp}_\infty(\boldsymbol{a}^*) + \frac{\epsilon}{2}$$

where the last line follows from the choice of $T = \frac{36\pi^2 n^2 p^2}{\epsilon^2}$. For $t_* = \mathrm{argmin}_t\, f_t(\boldsymbol{a}^{(t)})$ we therefore have

$$\langle M(\boldsymbol{a}^{(t^*)}), X^{(t^*)} \rangle = f_t(\boldsymbol{a}^{(t^*)}) \leq \mathrm{sdp}_\infty(\boldsymbol{a}^*) + \frac{\epsilon}{2}.$$

Since the value of $\mathrm{sdp}_\infty(\boldsymbol{a}^*)$ is bounded, the fact that $X^{(t^*)}$ is a multiplicative $(1+\frac{\epsilon}{10})$-approximation implies that $X^{(t^*)}$ is an additive $\frac{\epsilon}{2}$-approximation to $\mathrm{sdp}_\infty(\boldsymbol{a}^{(t^*)})$ (see Lemma A.4). Thus we conclude that

$$\mathrm{sdp}_\infty(\boldsymbol{a}^{(t^*)}) = \max_{X \succeq 0, X_{jj}=1} \langle M(\boldsymbol{a}^{(t^*)}), X \rangle \leq \langle M(\boldsymbol{a}^{(t^*)}), X^{(t^*)} \rangle + \frac{\epsilon}{2} \leq \mathrm{sdp}_\infty(\boldsymbol{a}^*) + \epsilon$$

i.e. $\boldsymbol{a}^{(t^*)}$ is an $\epsilon$-additive-approximation of $\min_{\boldsymbol{a}} \mathrm{sdp}_\infty(\boldsymbol{a})$.

To analyze the runtime note that each iteration requires approximately solving an instance of $\mathrm{sdp}_\infty(\boldsymbol{a}^{(t)})$, multiplying the solution $X^{(t)}$ by each vector $a_i^{(t)} - b_i^{(t)}$, and then rescaling by $\lambda^{(t)}$. First, the matrix $M(\boldsymbol{a})$ can be constructed in time $O(mn^{\omega-1})$ by grouping the vectors $a_i^{(t)} - b_i^{(t)}$ into $\frac{m}{n}$ blocks of size $n \times n$, and using fast matrix multiplication for each block. Next, using the fact that $M(\boldsymbol{a})$ is positive semi-definite, the approximate solution $X^{(t)}$ can be computed in time $\widetilde{O}(n^{7/2} \log(1/\epsilon))$ using the interior point SDP solver of [14] (see Lemma A.4 for details). Finally, we can group the vectors $a_i^{(t)} - b_i^{(t)}$ columnwise into $\frac{m}{n}$ matrices of size $n \times n$, and then use fast matrix multiplication to compute the product of each matrix with $X^{(t)}$. This results in a runtime of $O(mn^{\omega-1})$ where $\omega$ is the current matrix multiplication exponent. Finally, $\lambda^{(t)}$ can be computed in $O(mn)$ time. Putting it all together there are $O(\frac{p^2 n^2}{\epsilon^2})$ iterations each of which takes $\widetilde{O}(mn^{\omega-1} + n^{7/2} \log(1/\epsilon))$ time, given an upper bound $p \geq \mathrm{opt}(P)$. We can find an upper bound that is at most $2\,\mathrm{opt}(P)$ by starting with $p = \frac{1}{n}$ and repeatedly doubling at most $\log(n\,\mathrm{opt}(P)) = O(\log \rho)$ times. This yields the final iteration count of $(\frac{\rho^2 \log \rho}{\epsilon^2})$. $\qquad\square$

**$\ell_2$-norm Bounded Values.** We now turn to the setting where the data values are bounded in $\ell_2$-norm i.e. when $D = \{x \in \mathbb{R}^n \mid \|x\| \leq \sqrt{n}\}$. In this case the inner maximization in Definition 2.1 is equal to $n$ times the maximum eigenvalue of $M(\boldsymbol{a})$

$$\mathrm{sdp}_2(\boldsymbol{a}) = \max_{\|x\|_2 \leq \sqrt{n}} x^\top M(\boldsymbol{a}) x = n\|M(\boldsymbol{a})\|. \tag{3}$$

The choice of normalization is such that the second moment is one i.e. $\frac{1}{n}\sum_{j=1}^{n} x_j^2 = 1$. A subtle difference in the $\ell_2$-bounded setting is that $\mathrm{opt}(P)$ may not be bounded by a constant for certain pathological examples. This is unlike the $\ell_\infty$-bounded setting where choosing $\boldsymbol{a}$ to be the all zeros vector immediately gives an upper bound of $\mathrm{opt}(P) \leq 1$. However, if the target distribution is the full population mean (i.e. $b_i = \frac{1}{n}\mathbb{1}$ for all $i$), then setting $\boldsymbol{a}$ to all zeros and applying Cauchy-Schwarz yields

$$\mathrm{opt}(P) \leq \max_{\|x\|_2 \leq \sqrt{n}} \frac{1}{m} \sum_{i=1}^{m} \langle b_i, x \rangle^2 \leq \frac{1}{m} \sum_{i=1}^{m} \frac{1}{n} \cdot n = 1.$$

More generally, it only makes sense to compute estimators approximating the worst-case expected error when the achievable error goes to zero with $n$ i.e. $\mathrm{opt}(P) = o(1)$. Therefore, for the $\ell_2$-bounded setting we make the additional assumption that $\mathrm{opt}(P) \leq 1$. We now state our main theorem for the $\ell_2$-bounded setting.

**Theorem 2.4.** *Let $P$ be a sample-target distribution with $\mathrm{opt}(P) \leq 1$, $\rho = n \cdot \mathrm{opt}(P)$, and $\epsilon > 0$. There is an algorithm which computes a set of $m$ vectors $\boldsymbol{a}' = \{a_1', \ldots a_m'\}$ such that*

$$\mathrm{sdp}_2(\boldsymbol{a}') \leq \min_{\boldsymbol{a}} \mathrm{sdp}_2(\boldsymbol{a}) + \epsilon.$$

*The algorithm runs in $O\left(\frac{\rho^2 \log \rho}{\epsilon^2}\right)$ iterations and takes $O\left(\frac{\rho^2 \log \rho}{\epsilon^3} mn\right)$ time.*

If the optimal worst-case expected error is $\mathrm{opt}(P) = O(\frac{1}{n})$, then $O(\frac{1}{\epsilon^2})$ iterations suffices, and each iteration takes $O(\frac{mn}{\epsilon})$ time. The algorithm for the $\ell_2$-bounded case is also an application of online gradient descent, where in each step we solve an instance of $\mathrm{sdp}_2(\boldsymbol{a})$ and use it as a convex cost function. However, the fact that $\mathrm{sdp}_2(\boldsymbol{a})$ corresponds to computing the maximum eigenvalue of an $n \times n$ matrix leads to the improved runtime per iteration of $O(\frac{mn}{\epsilon})$.

---

**Algorithm 2:** Online gradient descent for minimizing $\mathrm{sdp}_2(\boldsymbol{a})$.

---

**Input:** A sample-target distribution $P$, accuracy parameter $\epsilon$, upper bound on optimum $p$.
**Output:** A sequence of $m$ vectors $\boldsymbol{a} = \{a_1, \ldots a_m\}$ with $a_i \in W_i$ for all $i$.

1   Let $r = \sqrt{mp}$
2   Let $\beta = \sum_{i=1}^{m} \|\Pi_{W_i^\perp} b_i\|^2$.
3   Let $a_i^{(1)} = \frac{1}{|A_i|} \mathbb{1}_{A_i}$ for all $i$.
4   **for** $t$ *from* $1$ *to* $\frac{36n^2p^2}{\epsilon^2}$ **do**
5      Let $\boldsymbol{a}^{(t)} = \{a_1^{(t)}, \ldots a_m^{(t)}\}$, and let $\eta_t = \frac{m}{n\sqrt{t}}$.
6      Let $X^{(t)} = x_t x_t^\top$, where $x_t$ is a $\left(1 + \frac{\epsilon}{10}\right)$-approximate solution to $\mathrm{sdp}_2(\boldsymbol{a}^{(t)})$.
7      $a_i^{(t+1)} \leftarrow a_i^{(t)} - \eta_t \frac{2}{m} \Pi_{W_i} X^{(t)}(a_i^{(t)} - b_i)$ for all $i$.
8      Set $\lambda^{(t)} = \min\left\{1, \sqrt{\frac{r^2 - \beta}{\sum_{i=1}^m \|a_i^{(t+1)} - \Pi_{W_i} b_i\|^2}}\right\}$.
9      $a_i^{(t+1)} \leftarrow \lambda^{(t)} a_i^{(t+1)} + (1 - \lambda^{(t)})\Pi_{W_i} b_i$.
10   **return** $\boldsymbol{a}^{(t^*)}$ where $t^* = \mathrm{argmin}_t \langle M(\boldsymbol{a}^{(t)}), X^{(t)} \rangle$.

---

To summarize briefly, the $\ell_2$-bounded algorithm can be essentially obtained by modifying Algorithm 1 by letting $X^{(t)} = x_t x_t^\top$ where $x_t$ is an $\frac{\epsilon}{2}$-additive-approximate eigenvector of $M(\boldsymbol{a}^{(t)})$. The improved running time follows from two points. First, the classical power method can compute the required eigenvector in $\frac{mn}{\epsilon}$ time. Second, computing the gradient requires multiplying $X^{(t)}(a_i^{(t)} - b_i)$ for all $i$, just as in Algorithm 1. However, since in the $\ell_2$-bounded case $X^{(t)} = vv^\top$ is rank one, each multiplication can be carried out in $O(n)$ time, for a total cost of $O(mn)$. The full details of the analysis can be found in Section B in the supplementary material.

## 3   The algorithm of Chen, Valiant and Valiant

At this point it is instructive to compare Algorithm 1 with the original SDP-based algorithm of [4]. In Appendix C of [4] the authors show how to approximate any sample-target distribution $P$ by drawing at most $m = \mathrm{poly}(\frac{n}{\epsilon})$ samples. Thus $m$ should be generally thought of as a large polynomial in $n$. The approach taken in [4] to approximate the optimal semilinear estimator is based on first replacing the inner maximization in Definition 2.1 with $\mathrm{sdp}_\infty(\boldsymbol{a})$. This only incurs an error of at most a factor of $\frac{\pi}{2}$, and additionally means that the problem is convex in $\boldsymbol{a} = \{a_1, \ldots a_m\}$ and linear in $X$. Thus the min-max theorem applies and the min and max can be exchanged, resulting in an inner minimization over $\boldsymbol{a}$ which can be solved explicitly. This gives rise to a somewhat complicated semidefinite program where the objective is a concave maximization depending on the inverse of the variable matrix.

The authors note that this SDP can be converted to a more standard SDP by using the Schur complement to re-express the matrix inverse. However this conversion increases the number of constraints in the SDP to at least $m$. The best known interior point solver [14] has runtime $\widetilde{O}(\sqrt{n}(mn^2 + m^\omega + n^\omega))$ for a general SDP with an $n$ dimensional PSD matrix variable and $m$ constraints. The dominant term in our setting is $\widetilde{O}(\sqrt{n}m^\omega)$. For example, if $m = O(n^3)$ then the runtime for the general SDP solver in [4] is $\widetilde{O}(n^{7.61})$, where we have used the fact that the current matrix multiplication exponent is approximately $\omega \approx 2.37$. The runtime of Algorithm 1 in this setting is $O(\rho^2 n^{4.37})$. Depending on the achievable worst-case expected error, $\rho$ can range from $O(1)$ to $O(n)$, so the runtime of Algorithm 1 ranges from $O(n^{4.37})$ to $O(n^{6.37})$, all of which are faster than $O(n^{7.61})$ for [4].

To summarize, after making the appropriate reductions, the algorithm of [4] requires solving an SDP with $O(m)$ constraints. In contrast, the SDP solved in each iteration of Algorithm 1 has exactly $n$

constraints. Therefore, given that SDP solvers have runtimes polynomial in the number of constraints, and that in our setting we should think of $m \gg n$, Algorithm 1 will tend to have a faster runtime even though it solves an SDP in each iteration.

## 4 A constant lower bound

In this section we identify a simple combinatorial condition that ensures the worst-case expected error of *any* estimator for the mean is at least a constant. Of course it is possible to construct trivial examples of sample-target distributions $P$ for which a constant lower bound holds. For example, consider a distribution $P$ where the samples $A_i \subseteq \{1, \ldots \frac{n}{2}\}$ and $B_i \subseteq \{\frac{n}{2} + 1, \ldots n\}$ for all $i$. In this case, given any estimator $f(x_A, A, B)$, a worst-case adversary is free to set the data values $x_j$ for $j \geq \frac{n}{2}$ arbitrarily in order to maximize $(f(x_{A_i}, A_i, B_i) - \text{mean}(x_{B_i}))^2$, as these values never appear as input to $f$. Indeed, since the estimator has zero probability of ever observing one of the data values $x_j$ for $j \geq \frac{n}{2}$, there is no hope of estimating the mean of these values under worst-case assumptions.

To go beyond the trivial case it makes sense to require that every data value has some non-negligible probability under $P$ of being included in a sample set $A$. We will show that even in this case, the worst-case expected error can be constant. In fact, the condition we identify can hold for distributions $P$ where each coordinate $i$ is equally likely to be included in $A$.

**Definition 4.1.** Let $\alpha > 0$ be a constant. A sample-target distribution $P$ is $\alpha$-non-expanding if there exists a subset $S \subseteq \{1, \ldots n\}$ such that for an $\alpha$-fraction of the pairs $(A_i, B_i)$ exactly one of the following holds:

1. $A_i \subseteq S$ and $B_i \cap S = \emptyset$

2. $A_i \cap S = \emptyset$ and $B_i \subseteq S$

For example, the definition is satisfied with $\alpha = 1$ by the distribution $P$ which half the time picks a uniform random subset $A$ from the first half of indices and $B$ from the second half, and half the time does the opposite. More subtle examples are possible where the subset $S$ is arbitrary and $\alpha < 1$, so that a constant fraction of sample and target sets can have arbitrary intersection with $S$.

**Theorem 4.2.** *Let $P$ be an $\alpha$-non-expanding sample-target distribution. For any estimator $f(x_A, A, B)$*

$$\max_{x : \|x\|_\infty = 1} \frac{1}{m} \sum_{i=1}^{m} (f(x_{A_i}, A_i, B_i) - \text{mean}(x_{B_i}))^2 \geq \frac{\alpha}{4}$$

*Proof.* Let $S$ be the subset provided by Definition 4.1. Without loss of generality we may assume that $A_i \subseteq S$ and $B_i \cap S = \emptyset$ at least for at least an $\frac{\alpha}{2}$ fraction of pairs $(A_i, B_i)$. Otherwise we could just switch the roles of $S$ and $\bar{S}$. Let $c$ be the median of $f(\mathbb{1}_{A_i}, A_i, B_i)$ on this $\frac{\alpha}{2}$ fraction. If $c$ is positive let $E$ be the at least $\frac{\alpha}{4}$ fraction of indices $i$ for which $f(\mathbb{1}_{A_i}, A_i, B_i) \geq c$. If $c$ is negative let $E$ be the at least $\frac{\alpha}{4}$ fraction of indices $i$ for which $f(\mathbb{1}_{A_i}, A_i, B_i) < c$. Set $x_j = 1$ for all $j \in S$ and $x_j = -\text{sign}(c)$ for all $j \notin S$. Since $A_i \subseteq S$ and $B_i \cap S = \emptyset$ for all $i \in E$ we have

$$\frac{1}{m} \sum_{i=1}^{m} (f(x_{A_i}, A_i, B_i) - \text{mean}(x_{B_i}))^2 \geq \frac{1}{m} \sum_{i \in E} (f(x_{A_i}, A_i, B_i) - \text{mean}(x_{B_i}))^2$$

$$= \frac{1}{m} \sum_{i \in E} (f(\mathbb{1}_{A_i}, A_i, B_i) + \text{sign}(c))^2$$

$$\geq \frac{\alpha}{4} (c + \text{sign}(c))^2$$

$$\geq \frac{\alpha}{4}$$

where the final inequality follows from the fact $|c + \text{sign}(c)| \geq 1$ for all $c \in \mathbb{R}$. $\square$

## 5 Experimental Results

In this section we empirically evaluate Algorithm 2 in settings where data values are correlated to inclusion in a sample. In [4] the authors show that solutions to $\text{argmin}_{\boldsymbol{a}} \, \text{sdp}_\infty(\boldsymbol{a})$ clearly outperform

standard choices for estimators in such settings. However, their algorithm requires using the general purpose convex programming solver MOSEK [2] via the CVXPY package [9, 1], which can be quite slow and memory intensive. We implement Algorithm 2 in Python, and run the algorithm on the same synthetic datasets as those in [4]. For the algorithm of [4] we use the publicly available code at `https://github.com/justc2/worst-case-randomly-collected`. We find that Algorithm 2 has comparable empirical performance to the original algorithm of [4], while being simpler and more computationally efficient. For more details of the experiments please see the supplementary material.

**Importance Sampling** Importance sampling, where elements are sampled independently but with different probabilities, is one of the simplest examples where data values may be correlated with sample membership. In this experiment the population size is $n = 50$ and the element $i$ is included in the sample with probability $0.1$ for $i \leq 25$ and probability $0.5$ for $i > 25$. We compare both Algorithm 2 and the $\mathrm{sdp}_\infty$-based algorithm from [4] with two standard estimators for this setting. The first estimator is *reweighting* which computes the weighted mean of the $x_i$, where the weight of each $x_i$ is the reciprocal of the probability that $x_i$ is included in the sample. The second is *subgroup estimation* which computes the sample means of $x_i$ for $i \leq 25$ and $x_i$ for $i > 25$ separately, and then averages these two sample means.

We evaluate the results on three synthetic datasets: (1) *Constant* where $x_i = 1$ for all $i$. (2) *Intergroup variance* where $x_i = 1$ for $i \leq 25$ and $x_i = -1$ for $i > 25$. (3) *Intragroup variance* where $x_i = 1$ for odd indices $i$ and $x_i = -1$ for even $i$. We also report worst-case $\ell_\infty$ and $\ell_2$ error by solving $\mathrm{sdp}_\infty(\boldsymbol{a})$ and $\mathrm{sdp}_2(\boldsymbol{a})$ for each estimator. The expected squared errors for each estimator and dataset in this experiment appear in Table 1. Note that on the three synthetic datasets the $\mathrm{sdp}_\infty$ algorithm from [4] and Algorithm 2 have essentially identical expected error, while the former has lower $\mathrm{sdp}_\infty$ error and the latter has lower $\mathrm{sdp}_2$ error as one might expect. Furthermore, the error on the synthetic datasets is consistently low for Algorithm 2, while the standard estimators each have large error for at least one setting where data values are correlated to sample membership.

| Data Values | Reweighting | Subgroup Estimation | $\mathrm{sdp}_\infty$ Alg. [4] | Algorithm 2 |
|---|---|---|---|---|
| Constant ($x_i = 1$) | 0.100 | 0.018 | 0.051 | 0.052 |
| Intergroup Variance | 0.100 | 0.018 | 0.053 | 0.052 |
| Intragroup Variance | 0.100 | 0.121 | 0.052 | 0.053 |
| Worst Case $\mathrm{sdp}_\infty$ | 0.101 | 0.122 | 0.053 | 0.062 |
| Worst Case $\mathrm{sdp}_2$ | 0.181 | 0.222 | 0.088 | 0.078 |

Table 1: Expected squared error for importance sampling experiment

**Snowball Sampling** In our snowball sampling experiment we randomly draw $n = 50$ points in the two dimensional unit square to be the population and let the target set $B$ be the entire population. We construct a sample by first picking a random starting point, and then iteratively adding to the sample two of the five nearest neighbors of each point added so far, until $k = 25$ points are included in the sample. We compare Algorithm 2 and the $\mathrm{sdp}_\infty$-based algorithm from [4] with the estimator that simply computes the sample mean. We evaluate these estimators on a dataset of spatially correlated values by setting $x_i$ equal to the sum of the two coordinates of the point in the unit square corresponding to element $i$. We also report worst-case $\ell_\infty$ and $\ell_2$ error by solving $\mathrm{sdp}_\infty(\boldsymbol{a})$ and $\mathrm{sdp}_2(\boldsymbol{a})$ for each estimator. The expected squared errors for each estimator and dataset in this experiment appear in Table 2. As in the previous example, we see that the $\mathrm{sdp}_\infty$ algorithm and Algorithm 2 have equal error on the synthetic dataset (both clearly outperforming the sample mean estimator), and each algorithm does better than the other on the worst-case data values it was designed to optimize.

| Data Values | Sample Mean | $\mathrm{sdp}_\infty$ Alg. [4] | Algorithm 2 |
|---|---|---|---|
| Spatially Correlate Values | 0.082 | 0.032 | 0.032 |
| Worst Case $\mathrm{sdp}_\infty$ | 0.690 | 0.135 | 0.153 |
| Worst Case $\mathrm{sdp}_2$ | 0.747 | 0.327 | 0.326 |

Table 2: Expected squared error for snowball sampling experiment

**Selective Prediction**   In the selective prediction experiment the population consists of $n = 32$ timesteps, and the goal is to predict some future target time window $\{t, \ldots, t + w\}$ given the sample consisting of the past $\{1, \ldots t\}$. Here $t < n$ is chosen uniformly at random, and $w$ is chosen uniformly from $\{1, 2, 4, 8, 16\}$. In this setting we compare the $\mathrm{sdp}_\infty$ algorithm from [4] and Algorithm 2 with the selective prediction estimator of [10, 18]. The selective prediction estimator simply computes the mean of the final $w$ elements of the sample $\{1, \ldots t\}$, and is known to achieve expected worst-case error $O(\frac{1}{\log n})$. We evaluate the worst-case $\ell_\infty$ and $\ell_2$ error of these by solving $\mathrm{sdp}_\infty(\boldsymbol{a})$ and $\mathrm{sdp}_2(\boldsymbol{a})$ and report the results in Table 3. As in the previous cases both the $\mathrm{sdp}_\infty$ algorithm from [4] and Algorithm 2 outperform the selective prediction estimator, and each of the two algorithms outperforms the other on the worst-case data values of the appropriate type.

| Data Values | Selective Prediction | $\mathrm{sdp}_\infty$ Alg. [4] | Algorithm 2 |
|---|---|---|---|
| Worst Case $\mathrm{sdp}_\infty$ | 1.208 | 0.498 | 0.620 |
| Worst Case $\mathrm{sdp}_2$ | 1.371 | 0.746 | 0.686 |

Table 3: Expected squared error for selective prediction experiment.

## Funding Disclosure

This work was partially supported by the Wallenberg AI, Autonomous Systems and Software Program (WASP) funded by the Knut and Alice Wallenberg Foundation.

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
