# A  Missing proofs for the $\ell_\infty$-bounded case

First we bound the diameter of the set of possible solutions for minimizing $\mathrm{sdp}_\infty(\boldsymbol{a})$.

**Lemma A.1.** *Let $P$ be a sample-target distribution. The optimum $\boldsymbol{a}^* = \mathrm{argmin}_{\boldsymbol{a}}\, \mathrm{sdp}_\infty(\boldsymbol{a})$ satisfies*

$$\frac{1}{m}\sum_{i=1}^{m}\|a_i^* - b_i\|^2 \leq \frac{\pi}{2}\,\mathrm{opt}(P)$$

*Proof.* Let $D = \{x \in \mathbb{R}^n \mid \|x\|_\infty \leq 1\}$ and $\boldsymbol{a} = \{a_1, \ldots a_m\}$. Recall that by the positive semidefinite Grothendieck inequality $\mathrm{sdp}_\infty(\boldsymbol{a})$ is a relaxation of the optimization problem $\max_{x\in D} x^\top M(\boldsymbol{a})x$, with value at most $\frac{\pi}{2}$ times larger. Thus

$$\min_{\boldsymbol{a}}\mathrm{sdp}_\infty(\boldsymbol{a}) \leq \frac{\pi}{2}\min_{\boldsymbol{a}}\max_{x\in D} x^\top M(\boldsymbol{a})x = \frac{\pi}{2}\,\mathrm{opt}(P)$$

Letting $\boldsymbol{a}^* = \mathrm{argmin}_{\boldsymbol{a}}\,\mathrm{sdp}_\infty(\boldsymbol{a})$ we then have

$$\frac{1}{m}\sum_{i=1}^{m}\|a_i^* - b_i\|^2 = \langle M(\boldsymbol{a}^*), I\rangle \leq \max_{X\succeq 0, X_{j,j}=1}\langle M(\boldsymbol{a}^*), X\rangle = \min_{\boldsymbol{a}}\mathrm{sdp}_\infty(\boldsymbol{a}) \leq \frac{\pi}{2}\,\mathrm{opt}(P)$$

as desired. $\qquad\square$

Next we compute a bound on the gradient of the cost functions used to minimize $\mathrm{sdp}_\infty(\boldsymbol{a})$ via online gradient descent.

**Lemma A.2.** *Let $\boldsymbol{a} = \{a_1, \ldots a_m\}$ with $a_i \in W_i$ for all $i$. Let $X \succeq 0$ be an $n \times n$ positive semidefinite matrix with $X_{jj} = 1$ for $j \in \{1, \ldots n\}$. Let $f(\boldsymbol{a}) = \langle M(\boldsymbol{a}), X\rangle$. For any $\boldsymbol{a}$ satisfying*

$$\sum_{i=1}^{m}\|a_i - b_i\|^2 \leq r^2$$

*we have*

$$\|\nabla f(\boldsymbol{a})\| \leq \frac{2nr}{m}$$

*Proof.* Recalling the definition of $M(\boldsymbol{a})$ we have

$$f(\boldsymbol{a}) = \langle M(\boldsymbol{a}), X\rangle = \frac{1}{m}\sum_{i=1}^{m}(a_i - b_i)^\top X(a_i - b_i).$$

Therefore the component of $\nabla f(\boldsymbol{a})$ corresponding to $a_i$ is given by

$$\nabla_{a_i} f(\boldsymbol{a}) = \frac{2}{m}\Pi_{W_i}X(a_i - b_i).$$

Thus we estimate the norm by

$$\|\nabla f(\boldsymbol{a})\|^2 = \frac{4}{m^2}\sum_{i=1}^{m}\|\Pi_{W_i}X(a_i - b_i)\|^2$$

$$\leq \frac{4}{m^2}\sum_{i=1}^{m}\|X(a_i - b_i)\|^2$$

$$\leq \frac{4}{m^2}\sum_{i=1}^{m}n^2\|(a_i - b_i)\|^2$$

where the last line follows from the fact that $\|X\| \leq \mathrm{Tr}\, X = n$. Plugging in the assumed bound on $\sum_{i=1}^{m}\|(a_i - b_i)\|^2$ yields the desired result. $\qquad\square$

The next lemma shows that final steps in Algorithm 1 correspond to projection onto $B_r(\boldsymbol{b})$, the ball of radius $r$ centered at $\boldsymbol{b}$. In particular, since $\boldsymbol{a} = \{a_1, \ldots a_m\}$ is restricted to the subspace where $a_i \in W_i$ for each $i$, the projection occurs within this subspace.

**Lemma A.3.** *For $\boldsymbol{a} = \{a_1, \ldots a_m\}$ let $W \subseteq \mathbb{R}^{nm}$ be the subspace where $a_i \in W_i$ for all $i$. Let $\beta = \sum_{i=1}^m \|\Pi_{W_i^\perp} b_i\|^2$. For any $r$ such that $B_r(\boldsymbol{b}) \cap W \neq \emptyset$, the projection $\mathrm{Proj}(\boldsymbol{a})$ of $\boldsymbol{a}$ onto $B_r(\boldsymbol{b})$ restricted to the subspace $W$ is given by:*

$$\lambda = \min\left\{1, \sqrt{\frac{r^2 - \beta}{\sum_{i=1}^m \|a_i - \Pi_{W_i} b_i\|^2}}\right\}$$

$$\mathrm{Proj}(a_i) \leftarrow \lambda a_i + (1-\lambda)\Pi_{W_i} b_i.$$

*Proof.* Note first that $\boldsymbol{b} = \Pi_W \boldsymbol{b} + \Pi_{W^\perp} \boldsymbol{b}$ for any $\boldsymbol{b} \in \mathbb{R}^{nm}$. The squared distance from $\boldsymbol{b}$ to $W$ is given by $\beta = \sum_{i=1}^m \|\Pi_{W_i^\perp} b_i\|^2$. Thus if $B_r(\boldsymbol{b}) \cap W \neq \emptyset$ then $\beta \leq r^2$ and the definition of $\lambda$ in the lemma statement makes sense.

Next observe that since each $\Pi_{W_i^\perp} b_i$ is orthogonal to all vectors in $W_i$

$$\sum_{i=1}^m \|a_i - b_i\|^2 = \sum_{i=1}^m \|a_i - \Pi_{W_i} b_i\|^2 + \sum_{i=1}^m \|\Pi_{W_i^\perp} b_i\|^2$$

for any $\boldsymbol{a} \in W$. Thus the intersection $B_r(\boldsymbol{b}) \cap W$ is equal to those vectors $\boldsymbol{a} \in W$ such that $\sum_{i=1}^m \|a_i - \Pi_{W_i} b_i\|^2 \leq r^2 - \beta$. This is precisely the ball of radius $r^2 - \beta$ in $W$ centered at $\Pi_W \boldsymbol{b}$. Thus, for any $\boldsymbol{a}$ not already in this ball, the projection is given by moving a $\lambda$ fraction of the distance along the line from $va$ to $\Pi_W \boldsymbol{b}$, exactly as described in the lemma statement. $\square$

Next we show how the fast interior point SDP solver of [14] can be used to solve an instance of $\mathrm{sdp}_\infty$ to the accuracy required for the proof of Theorem 2.3.

**Lemma A.4.** *A positive semidefinite matrix $X$ satisfying $\mathrm{sdp}_\infty(\boldsymbol{a}) \leq \left(1 + \frac{\epsilon}{10}\right)\langle M(\boldsymbol{a}), X\rangle$ can be compute in time $\widetilde{O}(n^{7/2} \log(1/\epsilon))$. Furthermore, if $\langle M(\boldsymbol{a}), X\rangle \leq \mathrm{sdp}_\infty(\boldsymbol{a}^*) + \frac{\epsilon}{2}$ then the approximation is additive i.e. $\mathrm{sdp}_\infty(\boldsymbol{a}) \leq \langle M(\boldsymbol{a}), X\rangle + \frac{\epsilon}{2}$.*

*Proof.* For matrices $M, C_1, \ldots C_k \in \mathbb{R}^{n \times n}$ and $b_i \in \mathbb{R}$, the interior point solver of [14] solves SDPs of the form

$$\begin{aligned}
&\text{Maximize: } \langle M, X\rangle \\
&\text{subject to: } \langle C_i, X\rangle = b_i \\
&\qquad\qquad\quad X \succeq 0
\end{aligned} \tag{4}$$

to accuracy $(1 + \epsilon)$ in time $\widetilde{O}(\sqrt{n}(kn^2 + k^\omega + n^\omega)\log(1/\epsilon))$. In our case $k = n$ as there are $n$ constraints of the form $X_{ii} = 1$ which can be equivalently written as $\langle e_i e_i^\top, X\rangle = 1$. Thus the dominant term in the runtime is $\widetilde{O}(\sqrt{n}kn^2 \log(1/\epsilon)) = \widetilde{O}(n^{7/2} \log(1/\epsilon))$ as desired.

Running the solver with accuracy parameter $\frac{\epsilon}{10}$ yields a solution $X$ such that

$$\mathrm{sdp}_\infty(\boldsymbol{a}) \leq \left(1 + \frac{\epsilon}{10}\right)\langle M(\boldsymbol{a}), X\rangle. \tag{5}$$

Letting $\boldsymbol{a}^* = \mathrm{argmin}_{\boldsymbol{a}} \, \mathrm{sdp}_\infty(\boldsymbol{a})$ we have by the positive-semidefinite Grothendieck inequality that

$$\mathrm{sdp}_\infty(\boldsymbol{a}^*) \leq \frac{\pi}{2} \mathrm{opt}(P) \leq \pi. \tag{6}$$

Thus, if $\boldsymbol{a}$ satisfies $\langle M(\boldsymbol{a}), X\rangle \leq \mathrm{sdp}_\infty(\boldsymbol{a}^*) + \frac{\epsilon}{2}$, then by (5) and (6)

$$\mathrm{sdp}_\infty(\boldsymbol{a}) \leq \langle M(\boldsymbol{a}), X\rangle + \frac{\epsilon}{10}\left(\pi + \frac{\epsilon}{2}\right) \leq \langle M(\boldsymbol{a}), X\rangle + \frac{\epsilon}{2}.$$

$\square$

# B   Analysis of the $\ell_2$-bounded case

The analysis of Algorithm 2 follows a similar outline to that of Algorithm 1, but is simpler in several regards. We begin with a lemma bounding the diameter of the set of feasible solutions to $\mathrm{sdp}_2(\boldsymbol{a})$.

**Lemma B.1.** *Let $P$ be a sample-target distribution. The optimum $\boldsymbol{a}^* = \mathrm{argmin}_{\boldsymbol{a}}\, \mathrm{sdp}_2(\boldsymbol{a})$ satisfies*

$$\frac{1}{m}\sum_{i=1}^{m}\|a_i^* - b_i\|^2 \leq \mathrm{opt}(P)$$

*Proof.* Let $D = \{x \in \mathbb{R}^n \mid \|x\| \leq \sqrt{n}\}$ and $\boldsymbol{a} = \{a_1, \ldots a_m\}$. By the definition of $\mathrm{sdp}_2(\boldsymbol{a})$ we have

$$\min_{\boldsymbol{a}} \mathrm{sdp}_2(\boldsymbol{a}) = \min_{\boldsymbol{a}} \max_{x \in D} x^\top M(\boldsymbol{a}) x = \mathrm{opt}(P)$$

Letting $\boldsymbol{a}^* = \mathrm{argmin}_{\boldsymbol{a}}\, \mathrm{sdp}_2(\boldsymbol{a})$ we then have

$$\frac{1}{m}\sum_{i=1}^{m}\|a_i^* - b_i\|^2 = \mathrm{Tr}(M(\boldsymbol{a}^*)) \leq n\|M(\boldsymbol{a}^*)\| = \min_{\boldsymbol{a}} \mathrm{sdp}_2(\boldsymbol{a}) = \mathrm{opt}(P)$$

as desired. $\qquad\square$

Next we bound the norm of the gradient of the cost function used in Algorithm 2.

**Lemma B.2.** *Let $\boldsymbol{a} = \{a_1, \ldots a_m\}$ with $a_i \in W_i$ for all $i$. Let $X = xx^\top$ with $\|x\| \leq \sqrt{n}$. Let $f(\boldsymbol{a}) = \langle M(\boldsymbol{a}), X \rangle$. For any $\boldsymbol{a}$ satisfying*

$$\sum_{i=1}^{m}\|a_i - b_i\|^2 \leq r^2$$

*we have*

$$\|\nabla f(\boldsymbol{a})\| \leq \frac{2nr}{m}$$

*Proof.* Recalling the definition of $M(\boldsymbol{a})$ we have

$$f(\boldsymbol{a}) = \langle M(\boldsymbol{a}), X \rangle = \frac{1}{m}\sum_{i=1}^{m}(a_i - b_i)^\top X(a_i - b_i).$$

Therefore the component of $\nabla f(\boldsymbol{a})$ corresponding to $a_i$ is given by

$$\nabla_{a_i} f(\boldsymbol{a}) = \frac{2}{m}\Pi_{W_i} X(a_i - b_i).$$

Thus we estimate the norm by

$$\begin{aligned}
\|\nabla f(\boldsymbol{a})\|^2 &= \frac{4}{m^2}\sum_{i=1}^{m}\|\Pi_{W_i} X(a_i - b_i)\|^2 \\
&\leq \frac{4}{m^2}\sum_{i=1}^{m}\|X(a_i - b_i)\|^2 \\
&\leq \frac{4}{m^2}\sum_{i=1}^{m} n^2\|(a_i - b_i)\|^2
\end{aligned}$$

where the last line follows from the fact that

$$\|X\| = \|xx^\top\| = \|x\|^2 \leq n.$$

Plugging in the assumed bound on $\sum_{i=1}^{m}\|(a_i - b_i)\|^2$ yields the desired result. $\qquad\square$

Next we show that an approximate eigenvector can be computed with the desired accuracy.

**Lemma B.3.** *A vector $x$ satisfying* $\mathrm{sdp}_2(\boldsymbol{a}) \leq \left(1 + \frac{\epsilon}{10}\right) \langle M(\boldsymbol{a}), xx^\top \rangle$ *can be compute in time* $\widetilde{O}(\frac{mn}{\epsilon})$. *Furthermore, if* $\langle M(\boldsymbol{a}), xx^\top \rangle \leq \mathrm{sdp}_2(\boldsymbol{a}^*) + \frac{\epsilon}{2}$ *then the approximation is additive i.e.* $\mathrm{sdp}_2(\boldsymbol{a}) \leq \langle M(\boldsymbol{a}), xx^\top \rangle + \frac{\epsilon}{2}$.

*Proof.* Let $L$ denote the $m \times n$ matrix whose rows are equal to $(a_i - b_i)$ and note that $L^\top L = M(\boldsymbol{a})$. By the classical power method of [15] we can compute a vector $v$ such that $\left(1 + \frac{\epsilon}{10}\right) v^\top M(\boldsymbol{a})v \geq \|M(\boldsymbol{a})\|$. The runtime is bounded by $\widetilde{O}(\frac{1}{\epsilon})$ times the cost of multiplying a vector by the matrix $M(\boldsymbol{a})$. Since $M(\boldsymbol{a}) = L^\top L$, we can split each matrix-vector product into two steps, first multiply by $L$ then by $L^\top$, for a total runtime of $O(mn)$. Thus, $v$ can be computed in $\widetilde{O}(\frac{mn}{\epsilon})$ time.

Letting $x = \sqrt{n}v$, and $X = xx^\top$ we have

$$\mathrm{sdp}_2(\boldsymbol{a}) \leq \left(1 + \frac{\epsilon}{10}\right) \langle M(\boldsymbol{a}), X \rangle. \tag{7}$$

Letting $\boldsymbol{a}^* = \mathrm{argmin}_{\boldsymbol{a}} \, \mathrm{sdp}_\infty(\boldsymbol{a})$ we have that

$$\mathrm{sdp}_2(\boldsymbol{a}^*) \leq \mathrm{opt}(P) \leq 1. \tag{8}$$

Thus, if $\boldsymbol{a}$ satisfies $\langle M(\boldsymbol{a}), X \rangle \leq \mathrm{sdp}_2(\boldsymbol{a}^*) + \frac{\epsilon}{2}$, then by (7) and (8)

$$\mathrm{sdp}_2(\boldsymbol{a}) \leq \langle M(\boldsymbol{a}), X \rangle + \frac{\epsilon}{10} \left(1 + \frac{\epsilon}{2}\right) \leq \langle M(\boldsymbol{a}), X \rangle + \frac{\epsilon}{2}.$$

$\square$

We are now ready to give the full proof of Theorem 2.4.

*Proof of Theorem 2.4.* The main observation is that the algorithm is precisely online gradient descent where the convex cost function observed at each step is given by $f_t(\boldsymbol{a}) = \langle M(\boldsymbol{a}), X^{(t)} \rangle$. Indeed viewing $\boldsymbol{a}$ as a vector in $nm$ dimensions (one for each $a_i$) the component of the gradient $\nabla f_t(\boldsymbol{a})$ corresponding to $a_i$ is $\frac{2}{m} \Pi_{W_i} X^{(t)} a_i$. By Lemma B.1 the optimal solution $\boldsymbol{a}^*$ lies in $B_{r^*}(\boldsymbol{b})$, the $\ell_2$-ball of radius $r^* = \sqrt{\mathrm{opt}(P)}$ centered at $\boldsymbol{b} = \{b_1, \ldots, b_m\}$. Thus given an upper bound $p \geq \mathrm{opt}(P)$ it is sufficient to limit the search to $B_r(\boldsymbol{b})$ for $r = \sqrt{mp}$. The last lines of the loop are just projection onto this ball (Lemma A.3). Finally, for $\boldsymbol{a}$ in $B_r(\boldsymbol{b})$, Lemma B.2 gives $\|\nabla f_t(\boldsymbol{a})\| \leq \frac{2nr}{m}$.

Thus the algorithm is online gradient descent with feasible set diameter $D = 2r$ and gradients bounded by $G = \frac{2nr}{m}$. Thus by the textbook analysis of online gradient descent [11] we have

$$\frac{1}{T} \sum_{t=1}^T f_t(\boldsymbol{a}^{(t)}) - \min_{\boldsymbol{a}' \in B_r(\boldsymbol{b})} \frac{1}{T} \sum_{t=1}^T f_t(\boldsymbol{a}') \leq \frac{3GD}{2\sqrt{T}} = \frac{3np}{2\sqrt{T}}.$$

Letting $\boldsymbol{a}^* = \mathrm{argmin}_{\boldsymbol{a}} \, \mathrm{sdp}_2(\boldsymbol{a})$ (which we know is contained in $B_r(\boldsymbol{b})$) we have

$$\frac{1}{T} \sum_{t=1}^T f_t(\boldsymbol{a}^{(t)}) - \frac{1}{T} \sum_{t=1}^T f_t(\boldsymbol{a}^*) \leq \frac{3np}{2\sqrt{T}}.$$

Noting that $f_t(\boldsymbol{a}^*) = \langle M(\boldsymbol{a}^*), X^{(t)} \rangle \leq \mathrm{sdp}_2(\boldsymbol{a}^*)$ yields

$$\frac{1}{T} \sum_{t=1}^T f_t(\boldsymbol{a}^{(t)}) \leq \mathrm{sdp}_2(\boldsymbol{a}^*) + \frac{3np}{2\sqrt{T}} \leq \mathrm{sdp}_2(\boldsymbol{a}^*) + \frac{\epsilon}{2}$$

where the last line follows from the choice of $T = \frac{36n^2 p^2}{\epsilon^2}$. For $t_* = \mathrm{argmin}_t \, f_t(\boldsymbol{a}^{(t)})$ we therefore have

$$\langle M(\boldsymbol{a}^{(t^*)}), X^{(t^*)} \rangle = f_t(\boldsymbol{a}^{(t^*)}) \leq \mathrm{sdp}_2(\boldsymbol{a}^*) + \frac{\epsilon}{2}.$$

Thus by Lemma B.3 $X^{(t^*)}$ is an $\frac{\epsilon}{2}$-additive-approximate solution to $\mathrm{sdp}_2(\boldsymbol{a}^{(t^*)})$, and so we conclude

$$\mathrm{sdp}_2(\boldsymbol{a}^{(t^*)}) \leq \langle M(\boldsymbol{a}^{(t^*)}), X^{(t^*)} \rangle + \frac{\epsilon}{2} \leq \mathrm{sdp}_2(\boldsymbol{a}^*) + \epsilon$$

i.e. $\boldsymbol{a}^{(t^*)}$ is an $\epsilon$-additive-approximation of $\min_{\boldsymbol{a}} \mathrm{sdp}_2(\boldsymbol{a})$.

To analyze the runtime note that each iteration requires approximately solving an instance of $\mathrm{sdp}_2(\boldsymbol{a}^{(t)})$, multiplying the solution $X^{(t)}$ by each vector $a_i^{(t)} - b_i^{(t)}$, and then rescaling by $\lambda^{(t)}$. First, by Lemma B.3 the vector $x_t$ can be computed in time $\widetilde{O}(\frac{mn}{\epsilon})$. Further $X^{(t)}(a_i^{(t)} - b_i^{(t)}) = x_t \langle x_t, (a_i^{(t)} - b_i^{(t)}) \rangle$ can be computed in time $O(n)$ for each $i$, for a total runtime of $O(mn)$.

Finally, $\lambda^{(t)}$ can be computed in $O(mn)$ time. Putting it all together there are $O(\frac{p^2 n^2}{\epsilon^2})$ iterations each of which takes $\widetilde{O}(\frac{mn}{\epsilon})$ time, given an upper bound $p \geq \mathrm{opt}(P)$. We can find an upper bound that is at most $2\,\mathrm{opt}(P)$ by starting with $p = \frac{1}{n}$ and repeatedly doubling at most $\log(n\,\mathrm{opt}(P)) = O(\log \rho)$ times. This yields the final iteration count of $(\frac{\rho^2 \log \rho}{\epsilon^2})$. $\qquad\square$

## C  Additional Details on Experiments

We based our code for the experiments (especially for the algorithm from [4]) on the publicly available code at `https://github.com/justc2/worst-case-randomly-collected`. The code is available under the MIT License. For running Algorithm 2 in practice we found that $T = 1000$ iterations was more the sufficient to compute a good solution. Though this is not a particularly scientific comparison, we found that in practice running the code on a laptop with an Intel 8th generation Core i5 and 16GB of RAM that Algorithm 2 was significantly faster than the public code for the algorithm from [4]. For example, in the snowball sampling experiment Algorithm 2 took about 3 seconds, while the algorithm from [4] took approximately 100 seconds to compile the program description into the correct form (including the automatic Schur complement reduction described in the previous section), and approximately 30 seconds to numerically solve the resulting SDP.