# OpenReview forum: "Faster Algorithms and Constant Lower Bounds for the Worst-Case Expected Error"
_NeurIPS.cc/2021/Conference — NeurIPS 2021 Poster_

### Official Review · Reviewer_R91w · 2021-07-13

**Rating:** 4
**Confidence:** 3

**Summary:**

This work builds on a recent statistical estimation framework of Chen, Valiant,
and Valiant [NeurIPS`20] that leverages knowledge about how samples are
collected but makes no distributional assumptions on the data values. The
authors explore three important questions in this space:

(1) Is is possible to compute a semilinear estimator for the $\ell_\infty$ case
that achieves the same approximation quality as that of Chen-Valiant-Valiant,
while also emitting explicit running time bounds?

(2) Is there an algorithm to compute a high-quality semilinear estimator in the
$\ell_2$-bounded case?

(3) Are there simple properties of a data collection process that guarantees no
estimator can achieve sub-constant worst-case expected error?

The results answer all three questions affirmatively. The authors show that
online gradient descent can be used to design simpler SDP-based algorithms,
answering (1) and (2). They also quantify a combinatorial property of joint
sample-target distributions that, for any estimator, necessarily incur at least
a constant error. The paper then presents experiments based on those in
[Chen-Valiant-Valiant, NeurIPS`20] to demonstrate the effectiveness of their
gradient descent-based approach.

**Main Review:**

*Originality.* This work builds on a fairly new problem introduced by Chen,
Valiant, and Valiant in NeurIPS`20, whose aim is to study statistical
estimation when data values are correlated with sample membership. The
algorithms proposed in this paper consider an SDP relaxation of the objective
function and then use traditional online gradient descent-based methods to
achieve the same approximation factor as Chen-Valiant-Valiant for the
$\ell_\infty$ case, but with an explicit bound on the running time. The
multiplicative approximation factor in both cases is the result of the SDP
relaxation. The extension to the $\ell_2$ case, together with the simple but
novel constant lower bound in Section 3, are nice contributions towards the
originality of this work.

*Quality.* Overall, this work is a nice extension of Chen-Valiant-Valiant. The
main results are incremental in some sense, but the approach is cleaner and
allow for further generalization (e.g., the $\ell_2$-bounded case). The
experiments do not really stand out, but they also don't detract from the
quality of the paper.

*Clarity.* The paper is written pretty well. The introduction could do a better
job motivating the model, but the message that it generalizes importance
sampling, snowball sampling, selective prediction, etc. comes across well
enough. The algorithms and analysis are quite clear, but the paper could
benefit from setting up the joint sample-target distribution definition in more
detail.

*Significance.* The least compelling part of the paper, in my opinion, is the
model itself (introduced in previous works). Given the problem, the algorithms
presented seem like good approaches, even though they are mostly theoretical
and rely on solving SDPs.

*Typos / Suggestions*
- [41] Missing a ',' in the set of data values.
- [57] Missing space in "selective prediction[10, 17]"
- [95] Even though Definition 1 references [4], the definition is initially
  somewhat unclear / ambiguous to me. A small example would be helpful, or an
  explicit definition.
- [110] Typo: "psd" --> "PSD"
- [118] Nit: "Recall that the SDP algorithm..." -- the reader might never have
  read [4], and hence can't recall. I suggest just saying that "The SDP algorithm..."
- [207] Typo: "sdp" --> "SDP"
- [249] Suggestion: Italicize \alpha-non-expanding since its the term being defined.
- [347] Journal is missing for Drucker reference [10]

**Time Spent Reviewing:**

3

---

> ### Author Response · Authors · 2021-08-10
> **Author Response**
>
> **Given the problem, the algorithms presented seem like good approaches, even though they are mostly theoretical and rely on solving SDPs.**
>
> While Algorithm 1 utilizes an SDP solver, Algorithm 2 does not. Instead, in each iteration Algorithm 2 needs only to compute an approximate top eigenvector which can be done very efficiently in theory and practice by the power method. We view one of the main contributions of our work to be showing that Algorithm 2 achieves similar worst-case expected error to the SDP-based algorithms, while not relying on an SDP solver.

---

### Official Review · Reviewer_Lwob · 2021-07-16

**Rating:** 7
**Confidence:** 4

**Summary:**

This paper studies the recently introduced *worst-case expected error* framework of Chen-Valiant-Valiant [4]. In this setting the data values are worst case, but the statistician has knowledge of the data-generating process. The authors provide new efficient algorithms based on alternating minimization-maximization for estimating the mean in worst-case under the previously studied $\ell_\infty$ assumption [4] on the dataset as well as under a new $\ell_2$ assumption. The authors also provide a simple combinatorial condition denoting when the worst-case error is $\Omega(1)$, and thus consistent estimation is not possible in this setting. Experiments compare their $\ell_2$ algorithm with the algorithm proposed by [4]  on synthetic data in the settings of importance sampling, snowball sampling, and selective prediction.

**Limitations And Societal Impact:**

Minor issues and typos:
- p2 Line 73: replace "minimization" with "maximization"
- p3 Line 101: replace "say at least $O(n^2)$" with "say $\Omega(n^2)$"
- p3 Theorem 1.2: I found the informal statement hard to parse because of the confusing phrase "within a multiplicative $\pi/2$ and additive $\varepsilon$ error". Please rephrase this
- p5 Line 188: I believe that the gradient (without projection) is actually $\frac{2}{m} X^{(t)} (a_i - b_i)$ and that the update in Line 7 of Algorithm 1 represents the projection of this gradient onto $W_i$ (*ie* back onto the feasible set). Is that correct? There is a similar statement in the Appendix for the $\ell_2$ that should also be fixed if so.
- p7 Definition 3.1: Very minor, but the term "$\alpha$-non-expanding" is a bit unwieldy. I suggest the terminology "$\alpha$-separated".


**Main Review:**

**Originality**: The authors make an interesting connection between the worst-case expected error objective and online optimization that enables them to introduce efficient algorithms with explicit run-time bounds. The authors also introduce a new $\ell_2$ assumption on the data that may be useful in some applications. The constant lower bounds are also new and shed further light on the degree of pessimism of this framework; in addition they apply to *any* estimator, not just the semilinear ones that were originally studied. One weakness along these lines is that there are no new examples given where this framework can be applied; the three examples studied were already introduced in [4]. The existing literature is well-described.

**Quality**: I have not checked the details in the appendix, but the arguments in the main text appear to be correct and use standard tools (albeit with clever application). The results are all interesting and provide a better understanding of this estimation framework as well as improved and explicit computational guarantees, improving on [4] where the bounds are polynomial-time but not explicit. I was a bit disappointed that Algorithm 1 was not evaluated empirically against the algorithm in [4]. This makes me concerned that the explicit bounds are still quite large (requiring $\varepsilon^{-8}$ iterations) and do not necessarily imply that Algorithm 1 is more computationally efficient than [4]. It would strengthen the paper to clarify this. On the other hand, it is interesting how the $\ell_2$ algorithm performs similarly to the algorithm from [4] in most cases.

**Clarity**: The paper is well-written and enjoyable.

**Significance**: The worst-case error framework is interesting, new, and still not yet well-studied, so this line of results is a welcome contribution that could inspire further study. Along these lines, it would benefit the paper to have some discussion of open problems and future directions for studying this framework.

**Time Spent Reviewing:**

8

---

> ### Author Response · Authors · 2021-08-10
> **Author Response**
>
> **This makes me concerned that the explicit bounds are still quite large (requiring $\epsilon^{-8}$ iterations) and do not necessarily imply that Algorithm 1 is more computationally efficient than [4]. It would strengthen the paper to clarify this.**
>
> For a more detailed discussion of the runtime bounds of Algorithm 1 and their relationship to the algorithm of [4] please see our response to Reviewer 52Vh. As described there, we will try to clarify the relationship between Algorithm 1 and [4] in more detail in the body of the paper.
>
> As for the question of practical performance, we believe that the results in the paper indicate that, on practical examples, there is little downside to using Algorithm 2 to minimize the worst-case expected error. While there are some examples where the $\ell_2$ normalization is more natural than the $\ell_{\infty}$ normalization and vice versa, the empirical results in the paper suggest that there isn’t a big difference between the two choices as far as minimizing error is concerned. Thus, Algorithm 2 is the clear choice in practice, as it only requires an approximate top eigenvector computation in each iteration (which can be done very efficiently both in theory and practice by the power method).

---

> > ### Comment · Reviewer_Lwob · 2021-08-24
> > **Thanks for the reply**
> >
> > Thanks for the reply. Also your response to reviewer 52Vh was helpful for understanding the computational issues. The advantages of your Algorithm 1 over that of [4] are rather subtle, so I agree with reviewer 52Vh that it would be good to discuss this point further in the main text.

---

### Official Review · Reviewer_52Vh · 2021-07-19

**Rating:** 7
**Confidence:** 3

**Summary:**

The authors build on the work of Chen, Valiant and Valiant from NeurIPS 2020 on estimators minimizing the worst-case expected error by providing two new algorithms for the same problem formulation, but with explicit runtime guarantees. The first algorithm uses a standard semidefinite program within a gradient descent loop to arrive at a runtime that is (with some abusive simplification) $\tilde O(mn^5 \log n/\epsilon^6)$. The second relaxes the setting so that data values are bounded in $\ell_2$ norm (rather than $\ell_\infty$), and uses this new setting to reduce the runtime to $O(mn^3 \log n/\epsilon^3)$. As far as I can tell, these algorithms don't offer any guarantees on sample complexity, but the newly explicit runtime guarantee is (subject to questions below) an improvement. As an impossibility result, the authors provide a combinatorial condition on the sample-target distribution under which the worst-case expected error for all estimators is at least a constant. Finally, the authors verify their new algorithm experimentally, showing comparable performance to [4] but faster runtime.

**Limitations And Societal Impact:**

The paper is almost entirely theoretical in nature and I don't foresee any direct societal impact as a result of this work.

I felt some (non-social) limitations of this work were not addressed in this paper as well as they could have been, most notably the one about sample complexity I mentioned in the main review.

**Main Review:**

**Clarity:** I thought the description of the framework from [4] and this work's intended contributions are very clear and the authors deserve credit and appreciation for this. There is perhaps some lack of clarity about certain points of differentiation, which I ask more about below and potentially suggest that the authors address in the main paper (rather than only supplementary). But the general thrust of the work is clear. The authors choose notation judiciously to aid communication and succinctness. The algorithms as written are mostly clear. The proofs, without the supplementary material, give a good understanding of the tools used to arrive at the results. The authors also articulate well how Algorithm 2 differs from Algorithm 1, and what about the additional structure leads to the improvements.

**Quality:** I checked the first and last paragraphs of the proof of Theorem 2.3, which is what I would consider to be the main result of this paper. I did not thoroughly check lines 194–201, nor did I peruse the lemmas in the supplementary material, or [16] (which was cited for the SDP solver, see questions below). Otherwise, the proof seemed reasonable to me and I did not detect any errors or problems with it. I felt that the understanding from this result carried over comfortably to Theorem 2.4. I also felt that the result in Theorem 3.2 was probably sound, but I did have one sticking point to clarify:

_Clarification in the proof of Theorem 3.2._ I couldn't quite follow the second-to-last step of the proof of Theorem 3.2 (line 264). I would have assumed that this follows directly from the definition of $c$ as $f(\mathbf{1} _ {A_i}, A_i, B_i) \ge c$. But I couldn't see how this would work when the median $c$ happens to be negative, since then $|f(\mathbf{1} _ {A_i}, A_i, B_i)|$ might be less than $|c|$.

As a counterexample for this step, imagine that $m = 8$ and $\alpha = 1$, so there are 4 samples in the category of interest, and consider $f(\mathbf{1}_{A_i}, A_i, B_i) = -i$ for $i = 1, \dots, 4$. Then I get $c = \frac{-5}{2}$ and

$$\frac18 \sum_{i \in E} [f(\mathbf{1}_{A_i}, A_i, B_i) + \mathrm{sign}(c)]^2 = \frac{13}{8} < \frac{49}{16} = \frac14 [c + \mathrm{sign}(c)]^2.$$

This probably doesn't disprove the theorem—if I'm not mistaken, I imagine this is probably quite easily resolved with some trivial definition swaps?

**Originality/Significance:** I hadn't come across [4] before reviewing this paper, so couldn't claim any knowledge of the area sufficient to fully assess originality and significance. However, it seems like [4] was the (or at least an) original work in this area, in which case this work seems clearly novel. I think the results, if the questions in this review are resolved, are very likely to be of interest to this community, because they make some promising headway into the open questions left by [4].

I'm not 100% sure on the following remark, but my understanding of this paper is that the authors claim that using online gradient descent can reduce the complexity of the SDP that is required to be solved as part of these algorithms (see questions below). If this is true, that seems like a significant insight and I would expect similarly interested researchers to build on these contributions.

**Questions about relationship with Chen, Valiant and Valient (2020) [4]:** I spent some time trying to understand precisely what the claimed improvements and sacrifices are relative to the algorithm provided in [4]. The description in Section C of the supplementary material was very helpful for this and I was a little surprised it didn't feature in the main paper. But a few questions remain that I hope the authors can address, perhaps by making minor revisions to the paper.

1. Since [4] made a claim about sample complexity, I tried to understand how the algorithms in this paper fare in terms of sample complexity. The authors didn't address this in the main paper, but argue in Section C that since Appendix C of [4] offers an approximation drawing $m = \mathrm{poly}(n, 1/\epsilon)$ samples (lines 472–473):
  >Thus $m$ should be generally thought of as a large polynomial in $n$. In contrast, both of our algorithms never use more than $O(mn)$ space […]

    Wouldn't, at a minimum, a similar argument that $m = \mathrm{poly}(n, 1/\epsilon)$ samples are necessary also apply to this work, in cases where $m$ _samples_ are taken, rather than just being the support size of $P$? What is it allows this work to treat $m$ as something less than $\mathrm{poly}(n, 1/\epsilon)$?

    Otherwise, my comment was going to be that I think it's fine for the authors not to offer sample complexity guarantees in this work (or to simply require that the entire sample-target distribution $P$ be specified, as [4] did in Theorem 3), but it should be mentioned explicitly as a limitation, trade-off or open question.

2. If I understood correctly, the algorithm in [4] is basically an SDP, [4] doesn't really address its runtime beyond saying it runs in polynomial time. In this work, the more explicit runtime guarantees are great, but the algorithm incorporates an SDP solver run once per iteration, for which they cite the runtime complexity provided by [16]. Why could this same solver from [16] and Lemma A.4 not be used on the SDP on which [4] relies?

    Again, Section C was of some assistance here. But the discussion merely suggested that the SDP in [4] is "somewhat complicated", and that the Schur complement conversion suggested in [4] requires $O(mn^2)$ memory. But memory use wasn't really discussed in the main paper, and I was still lost what this means in terms of runtime.

    Perhaps I'm asking the wrong question, but I think I was hoping for a discussion of why the SDP solver suggested in this work would not have worked as a much more straightforward improvement to [4]. That would have then offered some insight into why wrapping the SDP in online gradient descent can be an improvement over the single-shot SDP strategy of [4].

Depending on what the answers are to these questions, I may suggest addressing them in the main paper (even if only briefly) rather than only in supplementary material.

**Summary:** In summary, my assessment of this paper is generally positive, but conditioned on an understanding of the paper's core claims that I'm not totally certain about, and therefore dependent on the authors' assistance in addressing my questions above. I thank the authors in advance for their attention.

--------------------------

**Minor potential typos:** (for authors, not part of assessment)
- Line 188, is that meant to say $\frac{2}{m} \Pi_{W_i} X^{(t)} (a_i - b_i)$ rather than $\frac{2}{m} \Pi_{W_i} X^{(t)} a_i$?
- Line 207, was "sdp" meant to be capitalized?
- In the first two display math lines of line 264 (proof of Theorem 3.2), should the top limit in the summation sign $\sum_{i \in E}^m$ be removed, so that it's just $\sum_{i \in E}$?

--------------------------------
_Edit 02 Sep 2021: Fix whitespace/indentation, and change $\mathrm{poly}(n/\epsilon)$ to $\mathrm{poly}(n, 1/\epsilon)$._

**Time Spent Reviewing:**

7

---

> ### Author Response · Authors · 2021-08-10
> **Author Response**
>
> **Clarification in the proof of Theorem 3.2.**
> Thank you for catching this! You’re absolutely right, it is necessary to ensure that $f$ takes values with the same sign as $c$ on the set $E$. The corrected definition of the set $E$ in the proof of Theorem 3.2 is:
> If $c$ is positive, let $E$ be the set of indices $i$ for which $f(1_{A_i},A_i,B_i) \geq c$. If $c$ is negative, let $E$ be the set of indices $i$ for which $f(1_{A_i},A_i,B_i) < c$.
>
> **Questions about relationship with Chen, Valiant, and Valiant (2020)**
> What we were trying to do in our work was to separate the sample complexity concerns from the runtime analysis by treating the size $m$ of the sample-target distribution as a separate parameter. In particular, whatever support size/number of samples $m$ you need to accurately describe your sample target distribution, our analysis tells you the run time of the algorithm in terms of $m$.
>
> You are right that the same argument as that in [CVV’20] holds for the sample complexity in our work, so $m$ should also be thought of as a large polynomial in $n$ and $1/\epsilon$ in our work. The reason we mention this is not to argue that somehow $m$ is larger for [CVV’20], but rather to give an indication of how one should think about the runtimes we describe as a function of both $m$ and $n$.
>
> In particular, we mention that the algorithm of [CVV’20] eventually boils down to solving an SDP with $O(m)$ constraints. For a totally general SDP this takes $O(\sqrt{n}(mn^2 + m^{\omega} + n^{\omega}))$ time using the fastest known interior point method of [JKLPS’20]. The dominant term here in our setting is $O(\sqrt{n} m^{\omega})$. For example if $m = O(n^3)$ then the runtime of the general SDP solver applied to the SDP in [CVV’20] is $O(n^{7.61})$. Here we use the fact that the current matrix multiplication exponent is approximately $\omega = 2.37$. The runtime of Algorithm 1 in this setting is $O(\rho^2 n^{4.37})$. Depending on the achievable worst-case expected error, $\rho$ can range from $O(1)$ to $O(n)$ so the runtime of Algorithm 1 can range from $O(n^{4.37})$ to $O(n^{6.37})$, all of which are faster than $O(n^{7.61})$ for [CVV’20].
>
> To summarize at a higher level, and hopefully directly answer your question, once the appropriate reductions to a standard SDP have been done in [CVV’20] one obtains an SDP with $O(m)$ constraints. The SDP solved in each iteration of Algorithm 1 has exactly $n$ constraints. So given that SDP solvers have runtimes polynomial in the number of constraints, and that one should think of $m >> n$ in our setting, directly solving the SDP of [CVV’20] will generally not be competitive with Algorithm 1. In addition, the SDP solved in each iteration of Algorithm 1 has special structure which makes it amenable to somewhat faster positive SDP solvers, although this is less important than the difference in the number of constraints. Even if the same solver is used for both SDPs the fact that one has $O(m)$ constraints and the other has $n$ is the dominant factor.
>
> The above discussion covers the theoretical issues of runtime, but we found that a bigger obstacle for running these algorithms in practice is memory usage. The SDP of [CVV’20] uses $O(mn^2)$ space because it has $O(m)$ matrix variables of size $n \times n$. Even relatively small examples were able to fill up 16GB of RAM. Of course one could always buy more RAM, but the fact that Algorithm 2 only uses $O(mn)$ space was perhaps the most noticeable practical difference when trying to run examples on a laptop, which is why we mention it.
>
> We will definitely include more discussion of these issues of runtime comparison and sample complexity in the main body of the paper for the final version to improve the clarity and comparison with [CVV’20].
>
>
> [JKLPS’20]
> Haotian Jiang, Tarun Kathuria, Yin Tat Lee, Swati Padmanabhan, Zhao Song:
> A Faster Interior Point Method for Semidefinite Programming. FOCS 2020

---

> > ### Comment · Reviewer_52Vh · 2021-09-02
> > **Thank you!**
> >
> > With profuse apologies for the delay I wanted to thank the authors very much for their response to my clarification questions. The whole discussion was very helpful. In particular the paragraph about how the reductions of the [CVV'20] SDP to a standard SDP result in having $O(m)$ constraints whereas the SDP in this paper has $n$ constraints ("To summarize at a higher level, …") was especially enlightening for me. I appreciate the assurance that more discussion of this relationship will appear in a revised version. Thank you for taking the time!

---

### Official Review · Reviewer_FKv3 · 2021-07-20

**Rating:** 7
**Confidence:** 2

**Summary:**

This paper studies the _worst-case expected error framework_ of statistical estimation proposed by Chen, Valiant, and Valiant, where data values are worst-case, but the estimation algorithm exploits randomness in the data collection process.
In this worst-case expected error framework, there are $n$ data values $x_1$ to $x_n$ and a probability distribution $P$ over pairs of subsets $A, B \subseteq {1, \ldots, n}$, where the variables $x_A$ over the sample set $A$ are revealed, and we want to estimate the value of some fixed function $g(X_B)$ of the variables $x_B$ over the target set $B$.
Restricting attention to _semilinear_ estimators, which are linear combinations of the sample values whose weights could depend on the sample-target distribution $P$ or the indices $x_A$ and $x_B$ of the sample and target variables, Chen, Valiant, and Valiant gave a complicated semilinear estimator for the mean of the target set $B$ which is at most $\pi/2$ factor larger than the optimal semilinear estimator, when the data values are $\ell_\infty$-bounded.

This paper presents the following three results:
1. a simpler algorithm to compute a semilinear estimator within a multiplicative $\pi/2$ factor and an additive $\epsilon$ factor for $\ell_\infty$-bounded values, with explicit efficient runtime guarantees.
2. an efficient algorithm to compute a semilinear estimator within an additive $\epsilon$ error for $\ell_2$-bounded values.
3. a family of lower bound instances where all estimators (not necessarily semilinear) must have at least constant error, which generalizes the trivial lower bounds of disjoint $A$ and $B$.

**Limitations And Societal Impact:**

This theoretical-inclined paper does not discuss societal impact.

**Main Review:**

1. The main idea in the algorithms is to apply online gradient descent to solve an SDP relaxation arising from optimizing semilinear estimators.
2. This reader would appreciate more motivation of studying the generalized lower bounds, given that there are trivial lower bounds.

**Time Spent Reviewing:**

3

---

### Decision · Program_Chairs · 2021-09-27

**Decision:**

Accept (Poster)

**Comment:**

This paper continues a line of investigation initiated by Chen, Valiant, and Valiant in NeurIPS'20. That work proposed a new model of statistical estimation for worst-case data that is randomly collected and gave a polynomial time algorithm minimizes the expected error. The current work provides a significantly faster algorithm for this problem, matching the guarantees of the previous work, when the data is bounded in $\ell_{\infty}$-norm. Moreover, additional (new) results are obtained in the current work, e.g., for the setting that the data is bounded in $\ell_2$-norm. At the technical level, the proposed algorithms rely on some version of online gradient descent, as opposed to black-box convex optimization in the previous work. With the exception of one reviewer (who was unconvinced about the model itself, i.e., the prior NeurIPS'20 work), the reviewers ranked this paper above the acceptance threshold.